# Functional Characterization of *CsF3Ha* and Its Promoter in Response to Visible Light and Plant Growth Regulators in the Tea Plant

**DOI:** 10.3390/plants13020196

**Published:** 2024-01-11

**Authors:** Yan Bai, Rui Zou, Hongye Zhang, Jiaying Li, Tian Wu

**Affiliations:** 1School of Landscape Architecture and Horticulture Sciences, Southwest Forestry University, Kunming 650224, China; baiyan@swfu.edu.cn (Y.B.); 1499874271@swfu.edu.cn (H.Z.); lijy@swfu.edu.cn (J.L.); 2Qiannan Academy of Agricultural Sciences, Duyun 558000, China; zourui8556@dingtalk.com

**Keywords:** *CsF3Ha* gene, promoter, anthocyanin content, visible light, plant growth regulators, GUS

## Abstract

Flavanone 3-hydroxylase (F3H) catalyzes trihydroxyflavanone formation into dihydroflavonols in the anthocyanin biosynthesis pathway, serving as precursors for anthocyanin synthesis. To investigate the *CsF3Ha* promoter’s regulation in the ‘Zijuan’ tea plant, we cloned the *CsF3Ha* gene from this plant. It was up-regulated under various visible light conditions (blue, red, and ultraviolet (UV)) and using plant growth regulators (PGRs), including abscisic acid (ABA), gibberellic acid (GA_3_), salicylic acid (SA), ethephon, and methyl jasmonate (MeJA). The 1691 bp promoter sequence was cloned. The full-length promoter P1 (1691 bp) and its two deletion derivatives, P2 (890 bp) and P3 (467 bp), were fused with the *β-glucuronidase (GUS*) reporter gene, and were introduced into tobacco via *Agrobacterium*-mediated transformation. GUS staining, activity analysis, and relative expression showed that visible light and PGRs responded to promoter fragments. The anthocyanin content analysis revealed a significant increase due to visible light and PGRs. These findings suggest that diverse treatments indirectly enhance anthocyanin accumulation in ‘Zijuan’ tea plant leaves, establishing a foundation for further research on *CsF3Ha* promoter activity and its regulatory role in anthocyanin accumulation.

## 1. Introduction

The ‘Zijuan’ tea plant is classified as *Camellia sinensis* var. *assamica*, a specific variant rich in anthocyanins among the cultivated varieties of tea plants in the Yunnan large-leaf population. Typically, the anthocyanin content in the ‘Zijuan’ tea plant comprises 2.7–3.6% of dry matter [1], a notable contrast to the 0.01% in green tea. As ‘Zijuan’ leaves progress from a young to mature state, the anthocyanin accumulations induces a shift in leaf color, transitioning from purple to green due to reduced concentrations of flavonoids and anthocyanins. Anthocyanins, as secondary metabolites, impart red, purple, and blue pigmentation to all tissues of higher plants [2]. Widely distributed in fruits and most flowers of higher plants, anthocyanins belong to a group of flavonoids recognized for their health-promoting properties [3,4,5], including antioxidative, anti-inflammatory, and anticancer capacities [6,7].

The flavonoid biosynthetic pathway via the phenylpropanoid pathway leading to anthocyanins is well-established. CHS, CHI, F3′H, FLS, and F3H function in the early stages, yielding flavonols and other compounds. In the later stages, DFR, ANS, and UGT play crucial roles [8,9]. CHS initiates flavonoid biosynthesis, while FLS, IFS, and FNS contribute to the accumulation of colorless flavonoids (flavone and flavonol glycosides) [10]. Flavanone 3-hydroxylase (F3H), a pivotal enzyme in the central pathway, catalyzes the hydroxylation of naringin and other flavonoids to form 3-hydroxyflavonols [11]. This catalyzes the formation of trihydroxyflavanones into dihydroflavonols in the anthocyanin biosynthesis pathway, serving as common precursors for anthocyanins, flavanols, and proanthocyanidins. Exploring ways to enhance F3H’s ability to catalyze substrate conversion is crucial for promoting anthocyanin accumulation.

Anthocyanin susceptibility to pH, light, PGRs, temperatures, and metal ions, among other abiotic factors, is well-established. While the genetic background primarily determines the occurrence of specific anthocyanins in plants, environmental factors can variably affect the concentration of these compounds [12,13]. In *CsF3H* transgenic *Arabidopsis thaliana* seeds, a significant increase in flavonol glycosides and oligomeric proanthocyanidins was observed, accompanied by a decrease in monocatechin derivatives [14]. Research demonstrated significant induction of *CsF3H* genes under different abiotic stresses, including UV light radiation, sucrose, and ABA [14]. However, limited reports exist on the effects of environmental and biological factors on the promoters of different structural genes. The factors responsible for increased anthocyanin content remain largely unexplored. Studies on the *CsF3H* promoter in *Camellia sinensis* need further research. To elucidate the regulation of F3H expression, identifying promoter sequences interacting with regulatory proteins is crucial. In this study, we aim to clone the *CsF3Ha* gene and its promoter from the ‘Zijuan’ tea plant, predicting promoter-acting elements. *Agrobacterium*-mediated techniques were used to study different promoter segments’ activity. The investigation seeks to unveil the primary induction factors regulating anthocyanin content, offering insights for subsequent exploration of transcription factors associated with this gene.

## 2. Results

### 2.1. Changes of the Anthocyanin Content in Tea Plant under Different Treatments

The anthocyanin content in tea plant leaves varied significantly (*p* < 0.05) under visible light conditions, surpassing levels observed under white light (Figure 1A). Specifically, anthocyanin content differed across blue, UV, red, and white light settings, reaching peaks of 9.80 mg/g under blue light and 6.27 mg/g under white light. The results showed that the blue light significantly increased anthocyanin content (*p* < 0.05), and blue light was the most effective method for enhancing anthocyanin levels.

Anthocyanin content responded differentially to various PGR conditions. A notable downward parabolic trend in anthocyanin content accompanied an increasing ABA concentration (Figure 1B). When the ABA concentration increased to 100 mg/L, the anthocyanin content peaked (12.01 mg/g). However, beyond this threshold, it gradually decreased. A similar trend in anthocyanin content was observed with GA_3_ (Figure 1C). Upon reaching 100 mg/L GA_3_ concentration, the anthocyanin content peaked. The anthocyanin content exhibited a gradual increase with increasing SA concentration, reaching the maximum of 11.72 mg/g at 300 mg/L. Conversely, at 500 mg/L SA concentration, it reached the lowest of 9.49 mg/g (Figure 1D). Ethephon concentration correlated with a downward parabolic shift in anthocyanin content (Figure 1E). The ethephon significantly (*p* < 0.05) increased anthocyanin content at ≤100 mg/L, reaching a peak of 11.72 mg/g at 60 mg/L ethephon, sharply declining at 100 mg/L. This concentration marked a critical value, distinguishing between increased and decreased anthocyanin content compared to the control. The anthocyanin content changed significantly (*p* < 0.05) with increasing MeJA concentrations (Figure 1F), with the highest content of 12.36 mg/g observed at 150 mg/L MeJA concentration.

In summary, irradiating tea plant leaves with blue, UV, and red light or spraying with 100 mg/L ABA, 100 mg/L GA_3_, 300 mg/L SA, 60 mg/L ethephon, and 150 mg/L MeJA enhanced anthocyanin accumulation. These PGR concentrations served as the basis for subsequent experiments in this study.

### 2.2. Analysis of the Quantitative Real-Time PCR for Candidate Genes

In leaves of the tea plant with varying tenderness (Figure 2G) levels during the same period, the relative expression of genes in the anthocyanin synthesis pathway differed. The bud of the tea plant exhibits a significantly higher expression of *F3Ha* (Figure 2A), *F3Hb* (Figure 2B), and *ANSa* (Figure 2E) compared to the first, second, and third leaves. Conversely, the first leaf showed significantly higher expressions of *DFRa* (Figure 2C), *F3Ha*, and *ANSb* (Figure 2F) compared to the second and third leaves. Notably, *DFRb* expression in the bud, first leaf, and second leaf was significantly lower than that in the third leaf (Figure 2D). In conclusion, the expression of three genes was elevated in the bud and the first leaf, with *F3Ha* demonstrating a significantly higher expression level in the bud and the first leaf than that of other genes. Therefore, *F3Ha* was selected as a candidate gene for subsequent experiments to explore its regulation of anthocyanin synthesis.

### 2.3. Characterization of CsF3Ha Gene and Phylogenetic Tree Analysis

The *CsF3Ha* gene (Genbank: MT355076), spanning 1107 bp, was cloned, featuring open reading frames encoding 368 amino acids. It exhibited 99% homology compared to other plants in the tea group. Further, a multiple alignment analysis of the CsF3H protein and F3Hs from various plants revealed highly conservative amino acid sequences among F3Hs (Figure 3A). The 2OG-FeⅡ-Oxy oxidase domain, part of the 2-ketoglutaric acid and Fe (Ⅱ)-dependent oxidase superfamily, included the C terminal of the α subunit of prolyl 4-hydroxylase. A phylogenetic tree analysis encompassed *F3H* genes from 32 plants with high similarity (Figure 3B). Notably, the *F3H* gene from the ‘Zijuan’ tea plant demonstrated high similarities with *F3H* gene from *Camellia nitidissima* and *Camellia chekiangoleosa*.

### 2.4. Expression Analysis of CsF3Ha Gene

*CsF3Ha* expression significantly (*p* < 0.05) differed under visible light conditions (Figure 4A). Concurrently, *CsF3Ha* expression in one bud and two leaves of tea plants subjected to different visible light was significantly (*p* < 0.05) higher than under white light, following the order of blue, red, UV, and white light, with its expression being insignificant under UV light and the highest under blue light. Furthermore, *CsF3Ha* expression under different PGRs was significantly (*p* < 0.05) higher than that under H_2_O, following the order of ABA, SA, MeJA, GA_3_, ethephon, H_2_O, with it being the highest under ABA (Figure 4B).

### 2.5. Cloning and Cis-Element Analysis of the CsF3Ha Promoter

The 1691 bp fragment upstream of the ATG start codon was obtained and was preliminarily designated as *CsF3Ha*’s full-length promoter. The *CsF3Ha* promoter, as per predictions, exhibited typical eukaryotic features, including potential TATA-box, CAAT-box, and TA-rich intensifier elements. Several vital regulatory elements within the *CsF3Ha* promoter were identified (Figure 5A), encompassing numerous PGR response elements and *cis*-elements involved in ABA (CATGCA-motif), MeJA (CGTCA-motif), GA_3_ (GARE-motif), SA (TCA-motif), and auxin (TGA-motif) responses. Numerous elements, notably those related to light response (BOX 4, I-BOX, GATABOX, especially the blue, white, or UV -10 PEHVPSBD), were detected within specific regions: −467 to +1 bp (light (GATABOX), SA, ABA, and GA_3_ response elements), −890 to +1 bp (light (GATABOX, I-BOX, BOX 4, -10 PEHVPSBD), SA, ABA, and GA_3_ response elements), and −1691 to +1 bp (light (GATABOX, I-BOX, BOX4, -10 PEHVPSBD) SA, ABA, GA_3_, and MeJA response elements). Stress-responsive elements, including the CATGTG-motif (dehydration reaction), E-box (defense signaling), and MYB-binding site (drought inducibility), were also identified. Transcription factors binding to hormone response elements further suggested the potential regulation of *CsF3Ha* expression in tea plants by PGRs (GA_3_, MeJA, ABA, SA) and abiotic stress (light, water, and metal ions) (Table 1). To assess the functionality of *pCsF3Ha*’s various fragments, we constructed the full-length promoter P1 (1691 bp) and its two deletion derivatives, P2 (890 bp) and P3 (467 bp). Each fragment was fused to the *GUS* gene in the vector, creating three *pCsF3Ha::GUS* constructs for further examination (Figure 5B).

### 2.6. Transgenic Tobacco Lines with Hygromycin Resistance

The hygromycin-resistant lines, namely *pP1::GUS* (1–87 lines), *pP2::GUS* (1–96 lines), and *pP3::GUS* (1–84 lines), were generated (Figure 6). Upon seed germination, resistant buds exhibited robust growth, while individual buds with a white crown were identified as not hygromycin-resistant, possibly indicating false positive resistance. In the medium, the transgenic tobacco’s resistance allowed for normal growth, and its phenotype did not significantly differ from that of wild-type tobacco. The hygromycin-resistant lines were propagated monthly for subsequent experiments.

### 2.7. Histochemical GUS Staining

The GUS staining, observed under visible light and PGRs, was prominent in the *pP1::GUS* line excluding under UV light (Figure 7A). In contrast, the blue area in other treatments was less extensive than in the *pP2::GUS* line. The leaves of the *pP2::GUS* line all displayed blue staining across all treatments, the blue area extending throughout the leaves excluding those treated with UV light and ABA (Figure 7B). The GUS staining in the *pP3::GUS* line, visible under visible light and PGRs excluding those treated under UV light (Figure 7C), was mainly at the blue area on the tip of the leaves. Based on the staining area, the order was *pP2::GUS*, *pP1::GUS*, and *pP3::GUS.*

### 2.8. GUS Activity Assay

The enzyme activity of the *pP1::GUS* line under visible light surpassed that under white light, excluding UV light (Figure 7D). The highest activity occurred under blue light. Enzyme activity under PGRs was significantly higher than under H_2_O, with the peak under SA, but with no significant difference under ABA (Figure 7E). Similarly, the *pP2::GUS* line exhibited significantly higher enzyme activity under visible light and PGRs compared to H_2_O, with the highest activity under blue light and SA (Figure 7F,G). For *pP3::GUS*, enzyme activity under visible light and PGRs was significantly higher than under H_2_O, with the highest activity under red light and ABA, and no significant difference under UV light (Figure 7H,I).

### 2.9. The GUS Gene’s Relative Expression

The relative expression of *GUS* in the *pP1::GUS* line under visible light, especially blue light, was significantly higher than under white light, excluding UV light (Figure 7J). The relative expression was significantly higher under treatment with PGRs than with H_2_O besides ABA, and the highest with SA (Figure 7K). For *pP2::GUS*, both visible light and PGRs resulted in significantly higher relative expression, with the highest under blue light and ethephon (Figure 7L,M). In contrast, *pP3::GUS* showed significantly higher relative expression under blue light, red light, and PGRs compared to H_2_O, with the highest expression under blue light and SA, and no significant difference under UV light (Figure 7N,O). In summary, the leaves of the transgenic tobacco expressing the *CsF3Ha-GUS* construct exhibited responsive behavior to both visible light and PGRs.

## 3. Discussion

### 3.1. The Key Cis-Elements of CsF3Ha Promoter

The *CsF3Ha* promoter’s key *cis*-acting elements included blue light, red light, UV light, ABA, GA_3_, SA, and MeJA. Among these, light was one of the most important environmental factors affecting plant anthocyanin biosynthesis [15]. Data showed that the *CsF3Ha* promoter was significantly increased under blue light, red light, and UV light (Figure 7), with the light response element (BOX 4, I-BOX, GATABOX, and -10 PEHVPSBD) identified within the *CsF3Ha* promoter (Figure 5A). While predictive analysis exclusively revealed these light response elements, there might be other potential elements influencing *CsF3Ha* promoter activity that eluded prediction. Both blue light and UV light induced anthocyanin content by activating anthocyanin biosynthesis genes [16,17]. PGRs also impacted anthocyanin biosynthesis [18], with PGRs treatments up-regulating the *CsF3Ha* promoter (Figure 7). This suggested widespread distribution of PGR-responsive elements within P1, P2, and P3, consistent with promoter prediction results (Figure 5A). Interestingly, MeJA-related elements were not predicted in P2 and P3, but data demonstrated that P2 and P3 transgenic plants respond to MeJA, suggesting their potential as MeJA-inducing elements. PGRs, including ABA [19], MeJA [20], GA_3_ [21], and ethephon [22], play a role in foliar anthocyanin accumulation [23,24]. Notably, ethephon-related elements were not predicted in *CsF3Ha* promoter prediction analysis. Nevertheless, we speculated that P1, P2, and P3 might harbor potential ethephon-inducible elements. While the precise identification of *cis*-acting element sequences remains elusive, the *CsF3Ha* promoter undeniably contained a rich array of *cis*-acting elements associated with blue light, red light, UV light, ABA, GA_3_, SA, and MeJA.

### 3.2. Differences of CsF3Ha Promoter the P1, P2 and P3

The most robust expression of *GUS* was detected in *pP1::GUS* and *pP3::GUS* lines under blue light and SA conditions. In contrast, the most potent GUS expression was observed in *pP2::GUS* lines under blue light and ethephon, warranting further investigation (Figure 7). Interestingly, *pP2::GUS* lines exhibited expression under UV light, whereas *pP1::GUS* and *pP3::GUS* lines did not, suggesting possible inhibition of *pP1::GUS* and *pP3::GUS* activity by specific *cis*-elements under UV light. Histochemical GUS staining and GUS activity analysis validated these findings. Differences in the induction intensity of various promoter segments under diverse light and PGR conditions might be linked to specific *cis*-elements’ distribution in their sequences [25]. Following ABA treatment, the *GUS* activity of *pP3::GUS* was strongly induced, indicating that the ABA response element could activate gene expression, promoting *CsF3Ha* transcription under ABA treatment. Our study aligns with earlier reports that *CsF3Ha* functions through an ABA-dependent pathway [26,27]. Similar results were observed under GA_3_, SA, ethephon, and MeJA treatments. These findings echo those in other plants, where small promoter segments, like the 300 bp *AtTCTP* gene promoter [28], the 262 bp *MdHB-1* promoter [29], and the 171 bp *SaBS* promoter [30], effectively directed gene expression. In Figure 7, certain GUS expression data deviated from the observed GUS activity and staining results. Notably, discrepancies were observed in the *pP1::GUS* line under ethephon and GA_3_ treatments, as well as in the *pP2::GUS* line under ABA and GA_3_ treatments and the *pP3::GUS* line under red light and blue light treatments. In particular, after the ABA treatment, the *GUS* activity of *pP1::GUS* showed no significant difference, while ABA induction on staining resulted in a blue tint. This incongruity can be attributed to a temporal and spatial gap between gene transcription and translation [31]. The response of *pP3::GUS* to GA_3_ was weak, potentially due to the prevalence of numerous GA_3_-inhibiting transcription factors in P3. Additionally, *GUS* gene expression declined after certain treatments for six hours. However, the protein might remain up-regulated for an extended period due to the delay between transcriptional induction and protein level increases during state transition [32]. The above results indicated that the regulation pattern of the *CsF3Ha* promoter was an intricate behavior exerting a significant cumulative induction effect on anthocyanins and requires further investigation.

### 3.3. Molecular Regulatory Mechanism of CsF3Ha Promoter on Anthocyanin Synthesis in the Tea Plant

In this study, the *CsF3Ha* gene expression was up-regulated under blue light, red light, ABA, GA_3_, SA, ethephon, and MeJA treatments compared to controls (Figure 4). The visible light and PGRs stimulated the interaction between transcription factors and promoters, activating *CsF3Ha* promoter activity and inducing anthocyanin accumulation. To understand *CsF3Ha* promoter’s multiple functions, we predicted its transcriptional regulators in the model plant *Nicotiana tabacum*. The MYB transcription factor might directly bind to the *CsF3Ha* promoter, participating in flavonoid biosynthesis (Table 2). CIRCADIAN CLOCK ASSOCIATED1 (CCA1), an MYB transcription factor, was found to directly bind to the *GI* promoter [33]. Transcription factors like MYB, bHLH, and WD40 could regulate sweet potato’s anthocyanin synthase gene expression and synthesis [34]. Both *AaORA* and *AaERF2* activated *ALDH1* promoter expression [35]. *MdERF114* directly bound to the GCC-box of *MdPRX63* promoter, activating its expression and resulting in lignin deposition in apple roots [36]. R2R3-MYB transcription factors *PAP1* and *PAP2* were dependent regulators of anthocyanin biosynthesis [23]. In *Arabidopsis thaliana*, the complex of R2R3-MYB, bHLH, and WD40 (TT2, TT8, and TTG1) proteins activated proanthocyanidin gene expression [37]. Moreover, the WRKY transcription factor was WRKY DNA binding protein induced by SA, a cluster of WRKY binding sites that act as negative regulatory elements for the inducible expression of *AtWRKY18* [38]. The MYB transcription factor might bind to PGR-responsive elements in the *CsF3Ha* promoter and *Arabidopsis AtMYB2* was found to act as a transcriptional activator in ABA signaling [39]. Thus, we speculated that visible light (blue, red, and UV light) and PGRs (ABA, GA_3_, SA, ethephon, and MeJA) could enhance *CsF3Ha* promoter activity, initiating gene expression and inducing anthocyanin accumulation. Further research was needed to identify the molecular regulation mechanism of transcription factors on anthocyanin synthesis in the tea plant.

## 4. Materials and Methods

### 4.1. Plant Materials and Growth Conditions

In this study, Dr. Zhang Chi from Zhejiang A&F University generously provided *Nicotiana tabacum* seeds. The tobacco seedlings were cultivated for 30 days in a growth chamber at 25 °C, following a day/night photoperiod of 16/8 h, serving as the basis for genetic transformation. ‘Zijuan’ tea plants were grown in the experimental fields of Southwest Forestry University in Kunming, Yunnan Province, China (102°10′–103°40′ E, 24°23′–26°33′ N). Anthocyanin content analysis was conducted using one bud and two leaves of the ‘Zijuan’ tea plant.

### 4.2. Visible Light and PGRs Treatments on the Tea Plant

One bud and two leaves of the tea plant underwent treatment with blue, red, and ultraviolet (UV) light, along with the following plant growth regulators (PGRs): abscisic acid (ABA), gibberellic acid (GA_3_), salicylic acid (SA), ethephon, and methyl jasmonate (MeJA). The samples were arranged flat in the culture frame, with 2–3 layers of wet filter paper at the base. This ensured that the lower stem segment made contact with the wet filter paper, preventing sample water loss from influencing the experimental results. After placement, the samples received 12 h of red (1.8 W/cm^2^, 625–740 nm), blue (1.8 W/cm^2^, 400–480 nm), and UV light (3 W/cm^2^, 200–400 nm) exposure. Simultaneously, the leaves were sprayed with various PGRs at different concentrations, quickly sealed to avoid mutual influence, and left for a 12 h treatment period. The culture frame was positioned at approximately 25 °C with a 16 h/d day/night photoperiod, maintaining humidity. The control group was sprayed with deionized water and cultured under white light (1.8 W/cm^2^, 400–760 nm). Subsequently, the labeled samples were immediately frozen in liquid nitrogen and stored at −80 °C. The light source utilized was the T5 lamp.

### 4.3. Assay of Anthocyanin Content

The anthocyanin content in tea plant samples was quantified using the spectrophotometric method previously described with modifications by Abdel-Aal [40]. Plant materials (1 g), quick-frozen and ground with liquid nitrogen in a mortar, were then adjusted to 50 mL volume with acidic ethanol. After centrifugation at 4 °C, 2000–2500 rpm for 5 min, 4 mL supernatant was transferred to a tube. Then, 6 mL acidic ethanol were added and it developed color for 30 min. Acidic ethanol served as a blank control for measuring *OD*_525_, and anthocyanin content was calculated based on absorption values. The concentration of anthocyanin were measured by the following formula: anthocyanin (mg/g) = [(A fluid to be checked—A control solution) × 101.83/(4 × sample weight)]. Finally, the anthocyanin content was expressed as the amount in a 1 g sample.

### 4.4. Screening of Candidate Genes

The cDNAs from the apical bud and the first, second, and third leaves under the bud served as templates. A working solution of 1–5 ng/μL concentration was added to the fluorescence quantitative plate per the reaction system. The 20.0 μL reaction system comprised 10.0 μL EvaGreen 2 × qPCR Master Mix, 0.6 µL forward and reverse primers each (Table 2), 2.0 µL DNA template (<100 ng), and ddH_2_O used to adjust to a final volume of 20.0 µL. After attachment of the membrane, centrifugation was performed at 2400 rpm at 4 °C for 3 min, and then amplified by qPCR. The relative expressions of *F3Ha, F3Hb, DFRa, DFRb, ANSa,* and *ANSb* were calculated using the 2^−ΔΔCt^ method following Zhao [41], with the *β-actin* gene as the reference.

### 4.5. Analysis of the Quantitative Real-Time PCR for CsF3Ha Gene

Total RNA from buds or leaves was extracted using the Hipure HP Plant RNA Mini Kit (Magen, Shanghai, China) following the manufacturer’s instructions. Subsequently, first-strand cDNA was synthesized via reverse transcription using the respective kit (TransGen Biotech, Beijing, China) according to the manufacturer’s guidelines. Gene structure was elucidated by comparing coding and genomic sequences of *CsF3Ha* sourced from the *Camellia sinensis* var. *sinensis* cultivar genomic database (XP-028107405.1). The primers were then designed based on the full-length cDNA of *CsF3Ha* and *β-actin* (Table 2). The relative expression of *CsF3Ha* was calculated using the 2^−ΔΔCt^ method, as detailed by Zhao [41].

### 4.6. Promoter Cloning and Sequence Analysis

The 5’ upstream genomic sequence was obtained from the genomic database. Specific primers for promoter isolation were designed based on the reference sequence P F/R (Table 2). Genomic DNA, extracted from tea plant leaves using the DNA secure Plant Kit (Tiangen, Beijing, China), served as the template. The *CsF3Ha* promoter region was PCR-amplified under the following conditions: initial denaturation at 94 °C for 5 min, followed by 35 cycles at 94 °C for 30 s, 56 °C for 30 s, and 72 °C for 1 min, with a final extension at 72 °C for 10 min. The PCR product underwent purification, cloning into the pMD18-T vector (TaKaRa, Dalian, China), and sequencing by Shanghai Sangon Biotech Co. Ltd. (Shanghai, China). PlantCARE was utilized to predict the *CsF3Ha* promoter’s *cis*-elements [42].

### 4.7. Vector Construction of CsF3Ha Promotor and Transformation

The bacterial liquid from the preceding step served as a template. Three promoter fragments, named P1, P2, and P3, were derived using different primers (P1 F/R, P2 F/R, and P3 F/R, incorporating *EcoRI* and *SalI* restriction sites) (Table 2). These fragments comprised the full-length promoter (1691 bp, denoted as P1), −890 to +1 bp (designated as P2), and −467 to +1 bp (labeled as P3). The products were cloned and sequenced, and the correct sequence clones were obtained. The plasmids of the correct clones were extracted, and digested with *EcoRI* and *SalI*. Subsequently, they were ligated into the binary expression vector pCAMBIA1300-GUS, which had been digested by *EcoRI* and *SalI*. This process yielded three expression vectors: P1-GUS, P2-GUS, and P3-GUS. These plasmids were transformed into EHA105 and subsequently transformed into tobacco plantlet leaves.

Using the *Agrobacterium*-mediated leaf disc method, the P1, P2, and P3 constructs were transformed into tobacco [25]. Transgenic plants were screened on MS medium containing 20 mg/L of hygromycin. The transgenic plants after one propagation cycle were utilized for further GUS activity analysis.

### 4.8. Histochemical GUS Staining

After one month of growth, three transgenic tobacco lines (P1, P2, and P3), exhibiting robust growth and normal rooting, were randomly chosen. The leaves of the one positive plants were treated with visible light (red, blue, and UV light) and PGRs (100 mg/L ABA, 100 mg/L GA_3_, 300 mg/L SA, 60 mg/L ethephon, and 150 mg/L MeJA) for a 2 h duration. Controls involved white light irradiation and spraying with deionized water. Histochemical staining, following Jefferson’s protocol [43], was conducted on the leaves, which were subsequently incubated in GUS staining buffer, including 50 mM sodium phosphate buffer (pH 7.2), 2 mM 5-Bromo-4chloro-indolyl-b-D-glucuronide (X-Gluc), 2 mM K_3_Fe[CN]_6_, 2 mM K_4_[Fe(CN)_6_]·3H_2_O, 0.1% Triton X-100, and 10 mM Na_2_EDTA·2H_2_O for a 12 h duration at 37 °C. Stained samples were then cleared with 70% ethanol, and observed visually.

### 4.9. GUS Activity Assay

The leaves of lines and treatment methods used the same method as the histochemical GUS staining. The leaves were ground in a mortar, with a sample mass (g) to 0.01 M PBS (pH 7.2–7.4) volume (mL) ratio of 1:9. After homogenization, the sample was centrifuged, and the supernatant was collected. GUS enzyme activity was assessed using the GUS enzyme activity kit (Plant GUS Elisa Kit, Shanghai, China), following standard instructions. The absorbance value at 450 nm, obtained post-reaction, was measured using an enzyme label meter. GUS activity in the sample was then calculated based on the standard curve.

### 4.10. The GUS Gene Relative Expression

The leaves of lines and treatment methods used the same method as the histochemical GUS staining. The total RNA was extracted under RNase-free conditions using the Trigol reagent (Invitrogen, Carlsbad, CA, USA). The cDNA fragment was then synthesized using the Tiangen Quant cDNA reverse transcription kit (Tiangen, Beijing, China). The RNA quality and integrity were assessed via 1% agarose gel electrophoresis. Table 2 presents the list of primers for the *GUS* gene and reference genes (EF1α F/R) used in qRT-PCR *GUS* gene expression was calculated using the same method for screening candidate genes.

### 4.11. Statistical Analysis

Each P1, P2, and P3 transgenic tobacco line underwent three replicates. Three biological replicates were used for each measurement. Statistical analysis employed IBM SPSS Statistics 25, Origin 2023, and GraphPad Prism 9, utilizing the Duncan’s multiple range test for examining significant differences between groups. One-way ANOVA was used for multiple comparisons. All data was presented as mean ± standard error of the mean. A *p* < 0.05 indicated significance.

## 5. Conclusions

In this study, the *CsF3Ha* promoter regions of P1, P2, and P3 were fused to the *GUS* reporter gene, and many *cis*-elements were predicted. We observed a significant induction of the *CsF3Ha* promoter by SA and blue light under diverse abiotic stresses. Expression analysis of the *CsF3Ha* gene and anthocyanin content assay substantiated this observation. Thus, we speculated that transcription factors WRKY and MYB could bind *CsF3Ha* promoter motifs, enhancing its activity, initiating gene expression, and thereby inducing anthocyanin accumulation (Figure 8). This study elucidated the expression characteristics of the *CsF3Ha* promoter in tea plants, thus contributing to a deeper understanding of the complex regulatory mechanisms governing *CsF3Ha* expression in anthocyanin synthesis and accumulation. The functionally validated *CsF3Ha* promoter identified in this study stands as a promising candidate for genetic engineering to regulate anthocyanin synthesis and accumulation.

## Figures and Tables

**Figure 1 plants-13-00196-f001:**
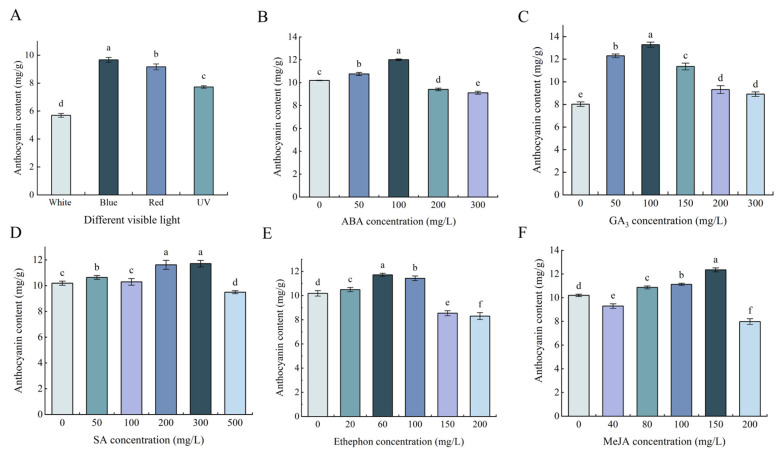
Effects of different treatments on anthocyanin content in tea plant leaves. (**A**): The change in anthocyanin content following irradiation with blue, UV, or red light. (**B**–**F**): Effects on anthocyanin content of different concentrations of ABA, GA_3_, SA, ethephon, and MeJA, respectively. Different small letters indicate significant differences at *p* < 0.05, and the same below.

**Figure 2 plants-13-00196-f002:**
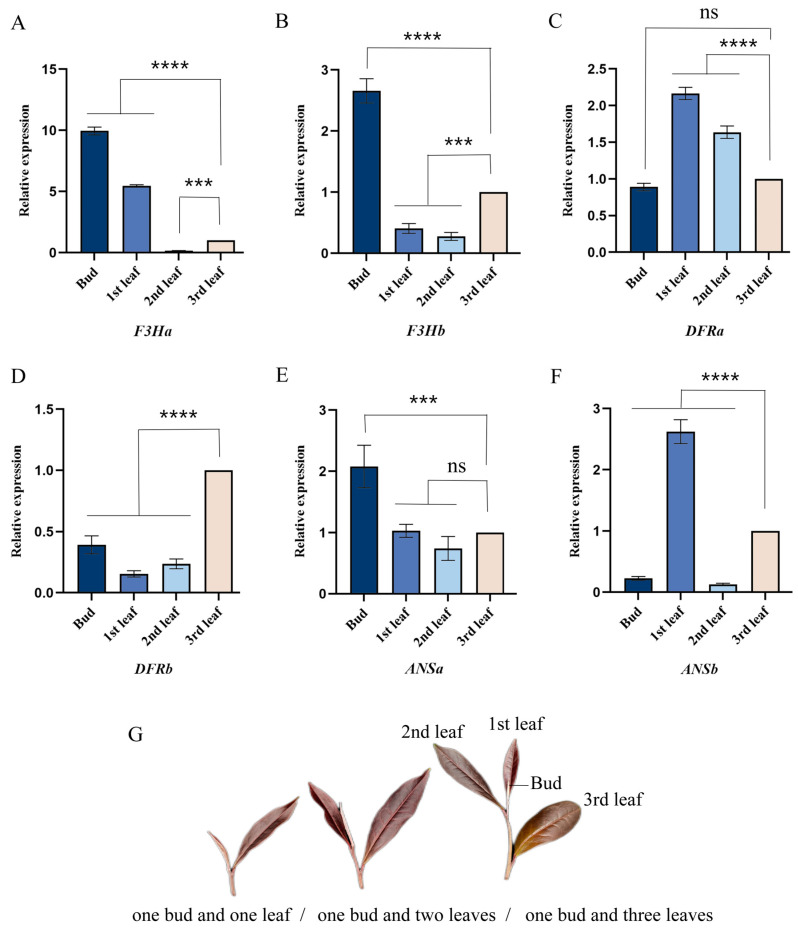
Relative expression of the genes in the anthocyanin synthesis pathway of tea plant leaves. (**A**–**F**): The relative expression of *F3Ha*, *F3Hb*, *DFRa*, *DFRb*, *ANSa*, and *ANSb* genes, respectively. (**G**): Schematic drawing of one bud and one leaf, one bud and two leaves, and one bud and three leaves. Note: the bar chart shows the mean of the three biological replicates, and the error bars are standard deviations, and the asterisks indicate statistical difference in one-way analysis of variance (ns *p* < 0.1234, *** *p* < 0.0002, **** *p* < 0.0001).

**Figure 3 plants-13-00196-f003:**
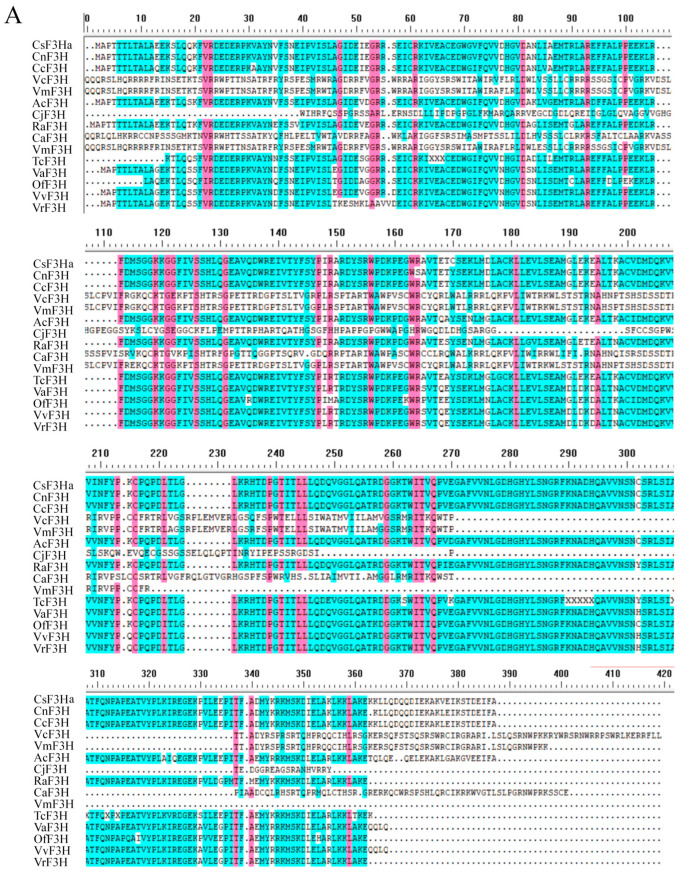
Multiple alignment of CsF3H protein and phylogenetic tree analysis of *CsF3Ha.* (**A**): Multiple alignment of CsF3H protein from other plants. (**B**): Phylogenetic tree analysis of *CsF3H* among multitudinous horticultural herbaceous and woody plants. *CsF3Ha* was highlighted with the red dot in the phylogenetic tree. Note: VaF3H (KP966099.1, *Vitis amurensis*), VvF3H (NM001281105.1, *Vitis vinifera*), VrF3H (KF040970.1, *Vitis rotundifolia*), SlF3H (KT954907.1, *Silene littorea*), MaF3H (ON055162.1, *Morus alba*), MnF3H (KF438044.1, *Morus notabilis*), OfF3H (MT495418.1, *Osmanthus fragrans*), OeF3H (XM023028959.1, *Olea europaea*), VwF3H (KJ463622.1, *Viola wittrockiana*), EmF3H (MW767838.1, *Euphorbia maculata*), VfF3H (KY656685.1, *Vernicia fordii*), TsF3H (MN906942.1, *Triadica sebifera*), CsF3Ha (MT355076, *Camellia sinensis*), CnF3H (HQ290517.1, *Camellia nitidissima*), CcF3H (HQ290517.1, *Camellia nitidissima*), MiF3H (KF513568.1, *Meconopsis integrifolia*), MsF3H (KF513571.1, *Meconopsis simplicifolia*), MbF3H (KF513559.1, *Meconopsis betonicifolia*), MrF3H (KF513570.1, *Meconopsis racemosa*), MhF3H (KF513566.1, *Meconopsis horridula*), MpF3H (KF513569.1, *Meconopsis pseudohorridula*), MdF3H (KF513565.1, *Meconopsis delavayi*), EuF3H (OK648437.1, *Erythronium umbilicatum*), RaF3H (OP018672.1, *Rhododendron agastum*), AcF3H (FJ542819.1, *Actinidia chinensis*), RpF3H (AB289594.1, *Rhododendron pulchrum*), MwF3H (MW373502.1, *Meconopsis wilsonii*), TcF3H (KU551927.1, *Thesium chinense*), ScF3H (AY332537.1, *Sinningia cardinalis*), ShF3H (LC322157.1, *Saintpaulia hybrid*), CaF3H (KY348830.1, *Camptotheca acuminata*), VcF3H (MH321462.1, *Vaccinium corymbosum*), VmF3H (MZ926696.1, *Vaccinium myrtillus*).

**Figure 4 plants-13-00196-f004:**
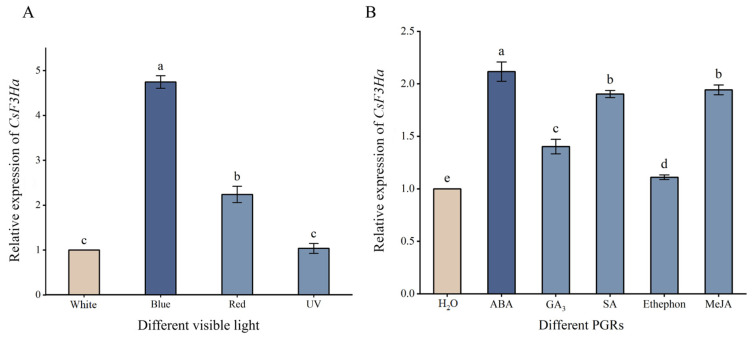
The expression of the *CsF3Ha* gene under different treatments. (**A**): Effects of visible light irradiation on *CsF3Ha* gene expression. (**B**): Effects of spraying different PGRs on *CsF3Ha* gene expression. Note: The samples were treated by red (1.8 W/cm^2^, 625–740 nm), blue (1.8 W/cm^2^, 400–480 nm), and UV (3 W/cm^2^, 200–400 nm) light, as well as 100 mg/L ABA, 100 mg/L GA_3_, 300 mg/L SA, 60 mg/L ethephon, and 150 mg/L MeJA for 12 h. The control group was sprayed with deionized water and cultured under white light (1.8 W/cm^2^, 400–760 nm) for 12 h.

**Figure 5 plants-13-00196-f005:**
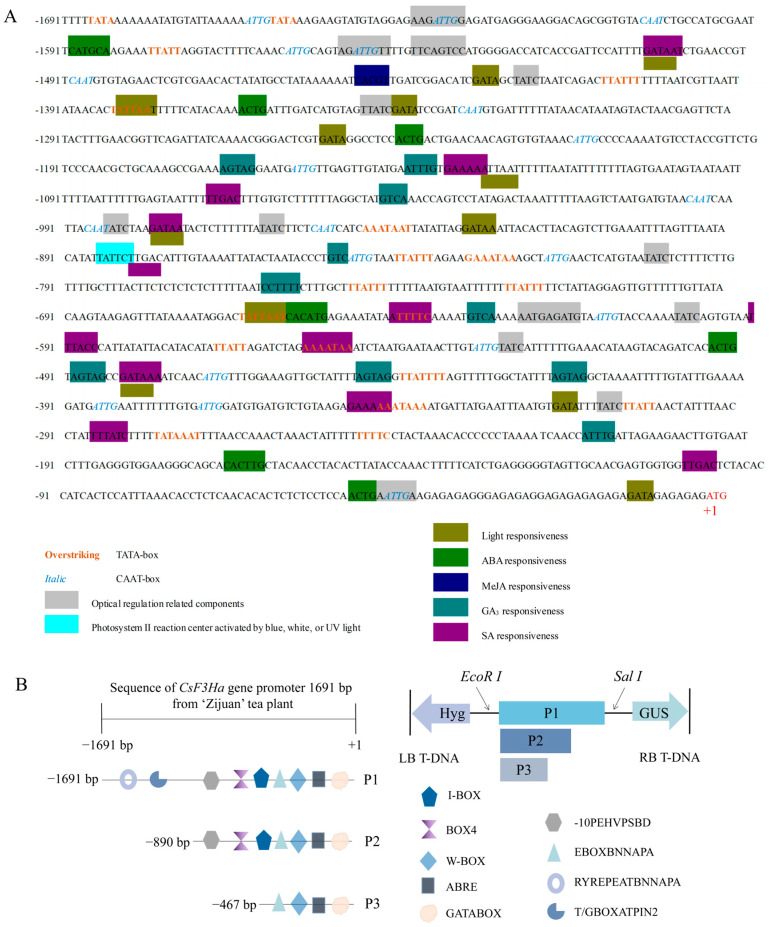
Predictive analysis of active elements in promoter sequence of the *CsF3Ha* gene. (**A**): Several potential *cis*-acting elements were predicted by the PLACE database, which were shaded colors and designated with the name of each of the motifs and constructed into an expression vector. The translation start codon ATG was displayed at the end of the promoter sequence, and the “A” in “ATG” was designated as “+1” position. (**B**): Diagram of the main *cis*-acting elements in the *CsF3Ha* promoter segments (P1, P2, and P3). The *cis*-elements associated with lights and PGRs were represented by different colored boxes.

**Figure 6 plants-13-00196-f006:**
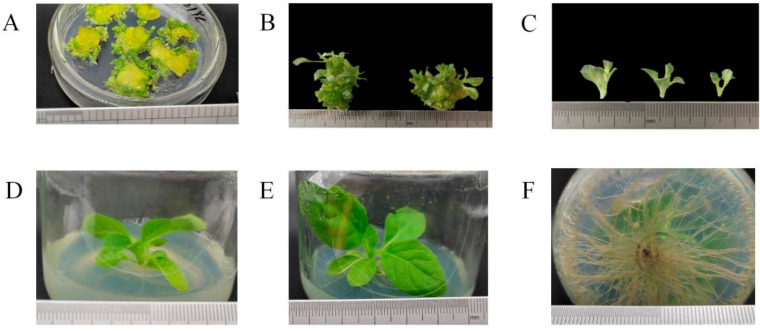
Transgenic tobacco in the media with hygromycin resistance. (**A**): Differentiated adventitious buds from the tobacco callus were screened for culture on 15th day. (**B**): Adventitious bud of hygromycin-resistant tobacco. (**C**): Adventitious buds cut from the callus. (**D**): Adventitious buds after 30 days. (**E**): Adventitious buds were induced during rooting. (**F**): Root of transgenic tobacco.

**Figure 7 plants-13-00196-f007:**
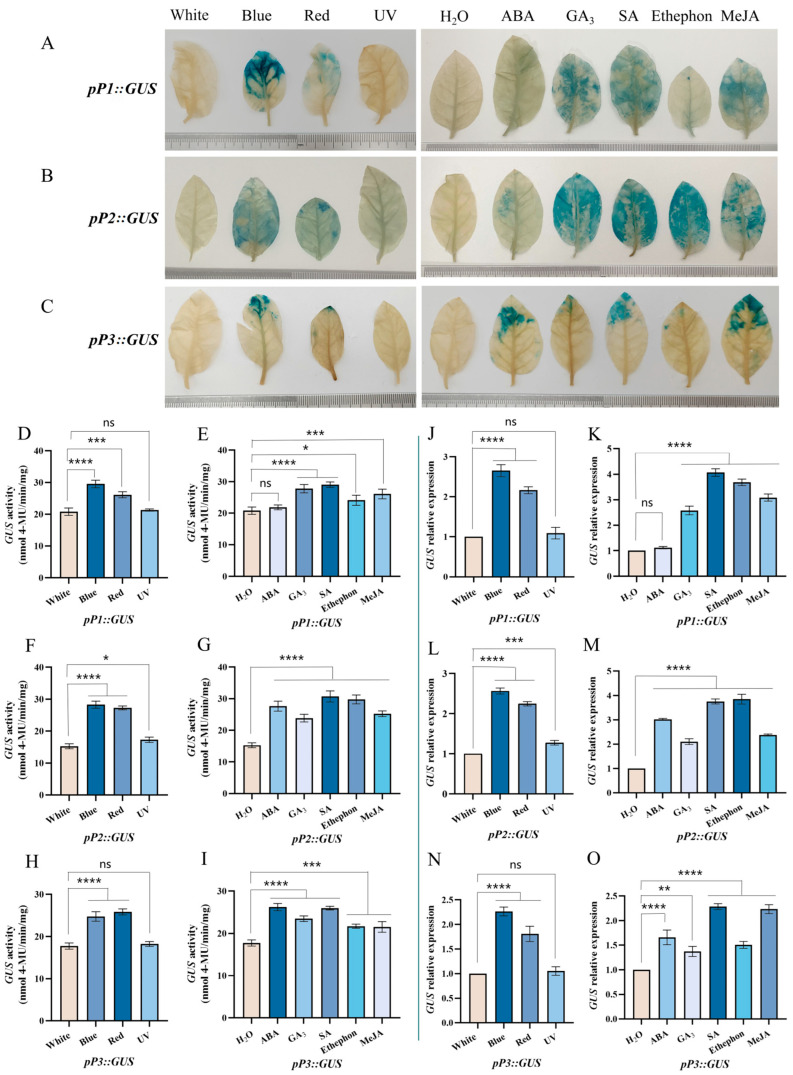
Histochemical GUS staining, GUS activity assay, and the *GUS* gene’s relative expression. (**A**–**C**): The GUS staining was detected under visible light and PGRs in the lines from *pP1::GUS*, *pP2::GUS*, and *pP3::GUS*. GUS activity assay under visible light and PGRs in the lines from *pP1::GUS* (**D**,**E**), *pP2::GUS* (**F**,**G**), and *pP3::GUS* (**H**,**I**). The *GUS* gene relative expression under visible light and PGRs in the lines from *pP1::GUS* (**J**,**K**), *pP2::GUS* (**L**,**M**), and *pP3::GUS* (**N**,**O**). The asterisks indicate statistical difference in one-way analysis of variance (ns *p* < 0.1234, * *p* < 0.0332, ** *p* < 0.0021, *** *p* < 0.0002, **** *p* < 0.0001).

**Figure 8 plants-13-00196-f008:**
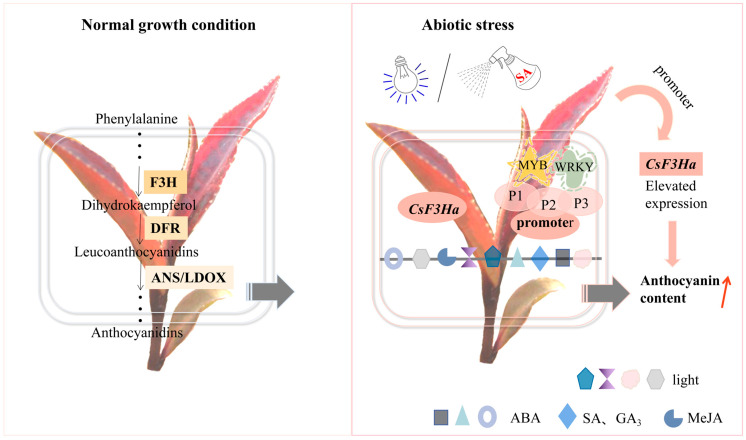
A model illustrating the role of the *CsF3Ha* promoter in anthocyanin accumulation of the ‘Zijuan’ tea plant.

**Table 1 plants-13-00196-t001:** Transcription-factor-related analysis of the *cis*-acting element of the *CsF3Ha* promoter.

*Cis*-Elements	Number	Function	Motif
W-BOX	10	transcription suppressor of the GA_3_ signaling pathway	TGAC
W-BOX	5	They were recognized specifically by salicylic acid (SA)-induced WRKY DNA-binding proteins	TTGAC
W-BOX	3	A novel WRKY transcription factor, SUSIBA2, participates in sugar signaling in barley by binding to the sugar-responsive elements of the iso1 promoter	TGACT
MYB PZM	1	*cis*-acting regulatory element involved in plant red pigment regulation	CCWACC
MYB PLANT	1	Promoters of phenylpropanoid biosynthetic genes, such as PAL, CHS, CHI, DFR	MACCWAMC
MYB CORE	3	*Cis*-acting regulatory element involved in flavonoid biosynthesis	CNGTTR
MYB1AT	2	MYB recognition site found in the promoters of the dehydration-responsive gene rd22 and many other genes in *Arabidopsis*	WAACCA

**Table 2 plants-13-00196-t002:** Primer sequences mentioned in this research.

Primer Name	Sequences (5′-3′)	Primer Name	Sequences (5′-3′)
F3HaF	CCGATTCTTGAAGAGCCAATCACG	F3HaR	TGAGAAGGCCAAGGTGGAAAT
F3HbF	TCAATAATGGAGGAACCAATCACC	F3HbR	GGAGTCCAAGCCAGTAGATGA
DFRaF	GAATCATTGAAACCTATCC	DFRaR	TAGAGAAAAGGGGATGTTGT
DFRbF	AAGGACTTGCCAGTTGTGT	DFRbR	CAGAAAACCCTGTCAATGGC
ANSaF	GTCTAGCAACAAAAGTCCTGTCG	ANSaR	GAGACGTCGGTGTGGGCTTCG
ANSbF	CGAGCCCTAACTACCAAGACG	ANSbR	GGGTTGAGGGCATTTTGGGT
β-actinF	GCCATCTTTGATTGGAATGG	β-actinR	GGTGCCACAACCTTGATCTT
PF	CCTGCCAACTGTCACCATACA	PR	TGAGGGACGAAGATGAACGT
P1F	CCGGAATTCTAAAATGTTGAGAATCG	P1R	GACGTCGACCTCTCAACTGAATCGAGCAA
P2F	CCGGAATTCAACCAAACTAAACTATT	P2R	GACGTCGACCTCTCTCTATCTCTCTCTCTCTCCTCTCTCCCTCTCTCTTCAA
P3F	CCGGAATTCCACGTTGATCGGACATCG	P3R	GACGTCGACGGTAGGACATTTTGGGGCAA
GUS-qF	GATCGCGAAAACTGTGGAAT	GUS-qR	TAATGAGTGACCGCATCGAA
EF1α-qF	TGGTTGTGACTTTTGGTCCCA	EF1α-qR	ACAAACCCACGCTTGAGATCC

The underlined were adding restriction sites.

## Data Availability

The data presented in this study are available on request from the corresponding author.

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
