# Peer review of "Functional Characterization of CsF3Ha and Its Promoter in Response to Visible Light and Plant Growth Regulators in the Tea Plant"

_plants, 2024, doi:10.3390/plants13020196_

Round 1

Reviewer 1 Report

Comments and Suggestions for Authors

The manuscript by Yan et al. uses biochemical, molecular, and bioinformatic approaches to characterize the promoter of F3H gene from tea plant. The authors have generated a solid amount of data and the conclusions are mostly justified by the results. However, some experiments and several details need to be clarified in the text or legends.

1. The article title is not clear. The characterization of CsF3Ha promoter is only in transgenic tobacco or also in tea plant?

1. In Results part.

Line 74-76: “Under different lights, the anthocyanin content was blue, UV, red, white light, and anthocyanin content was as high as 9.80 mg/g under blue light and 6.27 mg/g under white light”, what is the intensity and duration of different light conditions? How to measure the anthocyanin content? Please include the details either in Results or Materials and Methods.

Line 115 and 116: “To sum up, anthocyanin content was high in the apical bud and the first leaf under the bud of tea plant”, where is the results? Please include.

Line 118 and 119: “F3Ha was selected as a candidate gene in subsequent experiments to investigate its regulation of anthocyanin synthesis”, why select F3Ha but not F3Hb? What is the difference between F3Ha and F3Hb?

Figure 3: Please list the details from all the plants, e.g. GeneBank ID, species, …

Figure 4: Please include the treatment details, light intensity and duration, the concentration, …

Line 161 and 176: Why choose 1691, 890, and 467 bp? Please include more information or references.

Author Response

On behalf of my co-authors, we thank you very much for giving us an opportunity to revise our manuscript, we appreciate editor and reviewers very much for their positive and constructive comments and suggestions on our manuscript entitled "Functional Characterization of CsF3Ha Promoter in Response to Different Lights and PGRs in the Tea Plant" (plants-2781485).

We have studied reviewers' comments carefully and have made revision which marked in highlighted text in the manuscript. The main corrections in the paper and the responds to the reviewer's comments are the manuscript:

Reviewer 2 Report

Comments and Suggestions for Authors

In this manuscript Bai and co-authors present a study aimed to characterize the promoter of Camelia sinensis flavanone 3-hydroxylase (CsF3H) gene in response to visible light (white, blue and red) and UV as well as to plant growth regulators (PGRs). The response to various conditions was monitored in tobacco plants transformed with β-glucuronidase (GUS) gene under the control of CsF3H promoter (full length and two truncated versions). The authors also determined the variations in anthocyanin content under light (blue, red, UV) and PGRs exposure; the rationale for choosing the promoter of CsF3H gene for analysis is very well explained. The manuscript presents interesting data, but there are some issues that the authors need to address before the manuscript can be accepted for publication. Some issues are presented below.

-        The manuscript needs extensive revision of English, as some aspects are not fully clear. For example, lines 17-18 “ The GUS staining, the enzyme activity and the GUS gene relative expression showed that lights and PGRs could respond to promoter fragments”; lines 74-75 „Under different lights, the anthocyanin content was blue, UV, red, white light...”, and many others.

-        The abbreviations need to be defined the first time they appear in the text (e. g., PGRs, ABA, GA3, SA, etc.).

-        Figure 2, A-F: the title of vertical axis must be re-formulated. The same for Figure 7, D-O.

-        Irradiation conditions must be detailed (light source, wavelengths used).

Comments on the Quality of English Language

Some parts of the manuscript, especially the Abstract and the Introduction need extensive language revision.

Author Response

(The authors gave the same response as above.)

Round 2

Reviewer 2 Report

Comments and Suggestions for Authors

The authors responded to most reviewer's concerns, and the manuscript can be clearly considered for publication. 

There is still one issue: the authors may consider changing the phrase "Different Lights" in the title with "Visible Light", simply.  The phrase "Different Lights" is vague, and the fact that different domains of visible light wavelengths are used in the study is further specified within the text. 

Author Response

Dear Ms. Chen,

On behalf of my co-authors, we thank you very much for giving us an opportunity to revise our manuscript again, we appreciate editor and referee very much for their positive and constructive comments and suggestions on our manuscript (plants-2781485).

The referee suggested changing the “different lights” to “visible light”. We studied the referee’s comments carefully and made revision which marked in highlighted text in the manuscript.

Thanks again for the valuable revision comments of the referee and the hard work of the editorial department. Please do not hesitate to contact us if you have any questions.

Kind regards,

Prof. Tian Wu

E-Mail: [email protected]